# Exploring End-of-Life Care for Patients with Breast Cancer, Dementia or Heart Failure: A Register-Based Study of Individual and Institutional Factors

**DOI:** 10.3390/healthcare12090943

**Published:** 2024-05-04

**Authors:** Terje P. Hagen, Erika Zelko

**Affiliations:** 1Department of Health Management and Health Economics, University of Oslo, Blindern, P.O. Box 1072, 0316 Oslo, Norway; 2Institute of General Medicine, Johannes Kepler University, Altenberger Straße 69, 4040 Linz, Austria; erika.zelko@jku.au; 3Institute of Palliative Medicine, Medical Faculty, University Maribor, Slomskov trg 15, 2000 Maribor, Slovenia

**Keywords:** end-of-life care, palliative care, social and long-term care, Norway

## Abstract

Objective: To examine variations in end-of-life care for breast cancer, heart failure, and dementia patients. Data and methods: Data from four Norwegian health registries were linked using a personal identification number. Longitudinal trends over 365 days and the type of care on the final day of life were analyzed using descriptive techniques and logistic regression analysis. Results: Patients with dementia were more commonly placed in nursing homes than patients in the two other groups, while patients with heart failure and breast cancer were more frequently hospitalized than the dementia patients. Breast cancer and heart failure patients had a higher likelihood of dying at home than dementia patients. The higher the number of general practitioners, the higher was the probability of home-based end-of-life care for cancer patients, while an increasing non-physician healthcare workers increased the likelihood of home-based care for the other patient groups. Conclusions: Diagnoses, individual characteristics, and service availability are all associated with the place of death in end-of-life care. The higher the availability of health care services, the higher also is the probability of ending the life at home.

## 1. Introduction

End-of-life (EoL) care, though lacking a precise definition, generally refers to healthcare provided to individuals approaching death. This care encompasses ongoing treatment for the underlying disease, and palliative measures to manage symptoms and enhance the quality of life (QoL) [1]. The provision of EoL care typically involves a collaborative decision-making process, often supported by an established advanced care plan [1]. Emerging evidence from multiple studies suggests that initiating EoL care in a timely manner can bring about numerous benefits [2,3,4]. These include enhancing patients’ QoL, alleviating symptoms, and potentially reducing the unnecessary utilization of acute care services—which extends beyond just cancer patients [2,3,4]. Despite the beneficial effects of EoL and palliative care, global statistics indicate that only around 14% of patients in need receive palliative care [5]. Even in high-income countries, the results are comparable [6].

In Norway, where healthcare services are predominantly public and free, municipalities oversee primary health care, including primary palliative care and local emergency rooms (emergency primary healthcare clinics), while specialist healthcare is provided by four state-driven health regions, typically upon referral from primary care [7,8]. Palliative care is integrated into public health services, with specialist palliative care centers in hospitals staffed by at least one palliative care physician and one oncology nurse (ON). These specialists are available for consultation within hospitals and by primary care clinicians (general practitioners and ONs), who can also refer patients to them [8].

The geographical location of individuals at end-of-life has wide-ranging implications for healthcare delivery, costs, and, notably, for individuals’ preferences regarding care, particularly the realizations of desires to spend their final days at home [6,8]. Despite the widespread preference for home-based care at the end of life, the opportunity to do so is only available to a relatively small percentage of individuals, typically ranging from 10% to 30% in most countries [6,9,10,11,12,13,14], also including Norway [15].

Gomes and her team identified several essential conditions that are almost prerequisites for patients to have the option of spending their final days at home. These conditions include the patient’s own preference, the family’s preference, access to home palliative care, and the availability of district or community nursing [16]. In order to fulfill more individuals’ desires to receive end-of-life care at home and to comprehensively address their needs, Kellehear stresses that “end-of-life care is everyone’s business,” thereby extending responsibility beyond just families and healthcare services to encompass communities [17].

Research has consistently demonstrated a rise in healthcare service utilization during the final months of life [18,19,20,21]. However, there remains a need to fully understand the key variables that affect service utilization, including types of care at end-of-life. Our study endeavors to bridge this gap by examining disparities in service utilization during the twelve months prior to death among patients with breast cancer, dementia, and heart failure. Additionally, we aim to identify individual and institutional factors that influence the likelihood of patients dying at home. Of particular interest are potential associations between the supply of services at the local level such as GPs and other types of health personnel, and the odds of spending the last day of life at home.

## 2. Material and Methods

### 2.1. Inclusion Criteria and Data Sources

Utilizing data from the Norwegian Causes of Death Registry (NCDR), our analysis includes all patients who passed away in 2019 with underlying diagnoses of breast cancer (ICD-10 D05), dementia (ICD-10 F00–F03), or heart failure (ICD-10 I60, I61, I63, I64). By employing personal identification numbers obtained from the NCDR, we combined data from various registers, including the National Patient Register (a discharge register), the Municipal Patient and User Register, the Education Register, and KOSTRA—a register that describes municipal use of resources. The Directorate of Health oversees the first two registers, while Statistics Norway manages the latter two.

Data were collected for the period covering the last 365 days before the date of death for each patient, except for variables describing the patients’ co-morbidities where we collected data from the National Patent Register and the Municipal Patient and User Register for up to two years before the death date. All data were anonymized for the researchers.

### 2.2. Outcomes

The main outcomes were health service use the last 365 days before the death day (D0), including GP visits, home nursing, short- and long-term stays in municipal institutions (mainly nursing homes), as well as outpatient and inpatient stays in hospitals. Additionally, our analysis specifically investigated a binary variable indicating whether patients were at home (1) or in institutions (0), i.e., nursing home or hospital, on the day before their death (D-1). The reason for using D-1 as the time of measurement for ‘Dying at home’ was that services were not registered consistently on the death day.

### 2.3. Statistical Analyses

The characteristics of the cohorts were described by frequencies for categorical variables and by median for continuous variables.

To identify variables associated with ‘dying at home’ we performed a multivariate logistic regression analysis to estimate odds ratios (OR) and 96% confidence intervals (CI). We made separate analyses for the three cohorts that were defined by the causes of death with two groups of variables included, variables on patient and variables on municipal level. Variables on the individual level included gender, age categorized in 10-year age bands from 50 to 89 years and with patients below 50 and above 90 years in separate groups, marital status (indicator of informal care), education (primary, secondary, and higher education) and the number of comorbidities (0, 1–2, 3–4, 5 and above). The variables on the municipal level included the person years of GPs and caring personnel in total, both normalized by 10,000 inhabitants.

We registered data on 15 comorbidities (see Appendix A) from up to two years before the death day. Comorbidities were generated from the registration of both primary and secondary diagnoses and from both hospital inpatient and outpatient stays as well as consultations with GPs registered in the Municipal Patient and User Register.

Data management and analyses were conducted in SAS Studio 5.1 (SAS Institute Inc., Cary, NC, USA).

## 3. Results

### 3.1. Patient Population

In 2019, 606 patients succumbed to breast cancer, 2900 to dementia and 1415 to heart failure. Among breast cancer patients, the median age was 73.0 years, while for dementia patients it was 88.4 years and for heart failure patients it was 86.2 years (Table 1). When classified using 10-year age bands, the highest number of deaths occurred in the 70–79 age group, with 147 cases (24.3%) for the breast cancer patients. In the two other patient groups the highest numbers of death were in the age group 90 years and above.

The breast cancer patients also deviate from the two other groups, with a higher share being married, having fewer comorbidities, and higher education levels—naturally reflecting these patients’ younger age.

### 3.2. Service Use at End-of-Life

In the year leading up to their death, a significant proportion of breast cancer patients (64.0%) experienced at least one hospital admission, with 388 patients affected (Table 2). Furthermore, 556 patients (91.7%) received hospital treatments either as outpatients or during day stays. In contrast, the utilization of hospitals among patients with dementia in the year preceding death was notably lower, with only 10% admitted to hospitals and 33% receiving outpatient consultations or day stays. Dementia patients, however, were frequent users of nursing homes.

Breast cancer patients were found to make an extensive use of general practitioner (GP) services and frequently visited local emergency rooms (emergency primary healthcare clinics). Conversely, dementia patients had a different utilization pattern, with fewer individuals visiting GPs. It is important to note that while in nursing homes, patients receive medical services from an attending physician who is not part of the GP list patient system.

In terms of care profile, heart failure patients fell somewhere between the utilization patterns observed in cancer and dementia patients.

The dynamic changes in service utilization are further illustrated in Figure 1a–c, highlighting the use of services during each of the final 365 days before death among the patient groups. For all three patient groups, hospital stays remained relatively low but gradually increased during the last two months of life. Long-term stays in nursing homes were frequent and steadily increased among dementia patients, while they remained at a lower level among breast cancer patients. Notably, there was a significant increase in short-term stays in nursing homes among breast cancer patients during the last 2–3 months of life. Furthermore, for the breast cancer patients, a progressive increase in home nursing was observed until the last 4–6 weeks, followed by a decline in the number of recipients. This decrease was primarily due to patients being transferred to nursing homes, especially for short-term stays.

The proportion of patients residing at home without any of the aforementioned services gradually diminished, particularly for the breast cancer patients. This trend corresponded to the increasing number of patients receiving care in hospitals, nursing homes, and through home nursing services.

### 3.3. Factors Associated with Home Care at End-of-Life

The vast majority of patients (84%) who passed away from dementia did so in municipal institutions, mainly in nursing homes (Figure 2). Similarly, 57% of heart failure patients passed away in institutions. In contrast, for breast cancer patients, the distribution is almost equal, with 52% passing away in institutions and 48% at home.

The associations between patient characteristics and place of care during the last day before death are presented in Table 3. It is evident that, except for the dementia patients, strong associations exist between the variable describing age groups and staying at home on the last day of life, with the lowest age groups demonstrating significantly higher odds of staying at home compared to older age groups. While there are indications that the odds of staying at home on the last day of life increase with educational level, the relationship is only significant for the heart failure patients. Moreover, an increase in the number of comorbidities decreased the odds of staying at home, with significant effects observed for the heart failure patients.

The likelihood of cancer patients receiving end-of-life care at home is higher when there are more general practitioners available, while the likelihood of the other two patient groups receiving home care increases with the availability of non-physician healthcare workers. It is important to highlight that the notable disparities are observed between the group that has the least access to municipal care services and the other three categories. This implies that when access to home care is severely restricted, patients are more inclined to spend their remaining days away from home.

## 4. Discussion

We evaluated the utilization of healthcare services over the last twelve months of life among patients with breast cancer, dementia, and heart failure. The most significant differences were observed in hospitalizations and long-term care in nursing homes. Among the three patient groups, patients with dementia were most frequently placed in nursing homes, while the rate of hospitalization was highest among patients with heart failure and breast cancer. The breast cancer and heart failure patients had a higher likelihood of dying at home than the dementia patients. Furthermore, the availability of general practitioners increased the probability of end-of-life care at home for cancer patients, while the availability of non-physician healthcare workers increased the likelihood of staying at home at end-of-life for the other two patient groups.

Our research findings aligned with those of other authors [6,22,23]. Several studies note that dementia patients are less frequently hospitalized at EoL. The frequency of hospitalizations also decreases for other elderly patients with chronic conditions and those where palliative needs were recognized in a timely manner [24,25,26,27,28,29,30]. Diernberger and colleagues furthermore highlight the importance of the geographical environment, as they found that the frequency of hospitalization during the final stages of life among older adults living in rural areas was generally lower than for those living in urban areas. However, when hospitalization did occur, it tended to be of a longer duration [25]. Another important factor influencing the rate of hospitalization was the availability of beds in nursing homes [26,31]. This was further studied by Chu and colleagues, who found that the accessibility of care in nursing homes significantly reduced rehospitalizations, particularly for individuals in the advanced stage of dementia [26]. 

The utilization of healthcare services is influenced by numerous factors as analyzed by Williamson et al. [31]. We observed a lower utilization of healthcare services among higher-educated patients with heart failure but not for the other two groups of patients. Except for dementia patients, we observed that higher age increased the use of health care services, firmed by numerous other researchers [24,28,29,32,33,34,35,36,37]. Comorbidity had a weak negative impact on the utilization of healthcare services in our study, a finding echoed by other authors [29,38,39,40,41]. However, it might be that the effect of comorbidities interacts with age. The conclusion of a French study was that being younger and having comorbidities were identified as key factors associated with more intensive care and more frequent hospitalizations in the final stages of life [40].

Some researchers emphasize the need to consider care pathways of patients when assessing factors influencing the utilization of healthcare services in the final months of life [42]. As Norwegian researchers ascertain, age and access to informal care (marital status) are strong indicators of patients’ living arrangements and care [42]. The existence of local home-based palliative care and support are also associated with a greater likelihood of dying at home [43,44,45], a desire often expressed by many patients and their families [46,47,48,49]. Quinn and colleagues found that patients who received regionally organized, collaborative, home-based palliative care experienced a 48% reduced risk of hospital death compared to those receiving standard care. Noteworthy advantages of this approach included increased clinician home visits, postponed initial hospital admissions, shorter hospital stays, and more time spent at home [45]. Our study echoes this by finding that better access to formal care, be it either GPs or other health care workers, increased the odds of ending the life at home. Unfortunately, the frequency of palliative care for patients is lower than would be necessarily for enabling dying at home, especially for non-cancer patients [39,50,51].

A strength of our analysis is the use of data registries that cover the whole Norwegian population. Our sample could have been larger by including a longer time period, for example from 2019 to 2021. However, as the COVID-19 pandemic affected health care use from 2020, we decided not to do so.

## 5. Conclusions

Diagnoses, individual characteristics, and service availability are all associated with the place of death. The higher the availability of health care services, the higher also is the probability of ending life at home.

## Figures and Tables

**Figure 1 healthcare-12-00943-f001:**
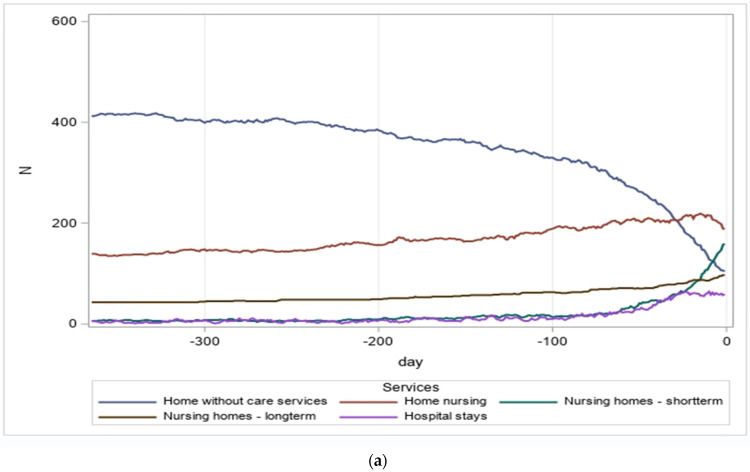
(**a**) Service use by day, last 365 days of life, breast cancer patients (N = 606); (**b**) service use by day, last 365 days of life, dementia patients (N = 2900); (**c**) service use by day, last 365 days of life, heart failure patients (N = 1415).

**Figure 2 healthcare-12-00943-f002:**
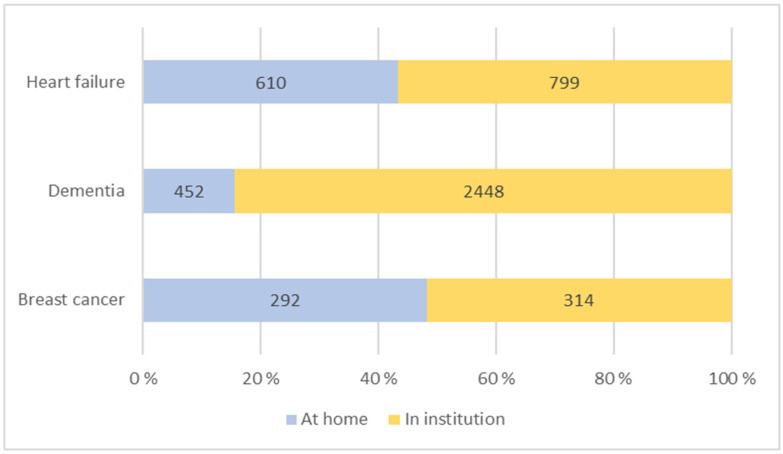
Number of patients at home or in institutions during the last day of life.

**Table 1 healthcare-12-00943-t001:** Patient characteristics.

		Breast Cancer	Dementia	Heart Failure
		N (%)	N (%)	N (%)
Total		606 (100)	2900	1415
Gender	Female	600 (99.0)	1972 (84.4)	837 (59.2)
	Male	6 (1.0)	452 (15.6)	578 (40.9)
Age	<50 years	44 (7.3)	0 (0.0)	11 (0.8)
	50–59 years	98 (16.2)	2 (0.0)	19 (1.4)
	60–69 years	115 (19.0	35 (1.2)	50 (3.6)
	70–79 years	147 (24.3)	287 (9.9)	186 (13.2)
	80–89 years	129 (21.3)	1175 (40.5)	460 (32.7)
	90 years≤	73 (12.1)	1402 (48.3)	683 (48.5)
	Median	73.0	88.4	86.2
Education	Primary	177 (29.2)	1279 (44.4)	635 (45.5)
	Secondary	271 (44.7)	1209 (42.0)	612 (43.8)
	Higher	152 (25.1)	390 (13.55)	150 (10.7)
	Missing	6	22	12
Marital status	Others *	358 (59.1)	2110 (72.8)	1030 (73.1)
	Married	248 (40.9)	790 (27.2)	379 (26.9)
Comorbidities	0	305 (50.3)	298 (10.3)	133 (9.4)
	1–2	196 (32.3)	1318 (45.5)	443 (31.4)
	3–4	74 (12.2)	871 (30.0)	442 (31.4)
	5 or more	31 (5.1)	413 (14.2)	391 (27.8)
General Practitioners (GPs) per 10,000 inhabitants	<10.1	154 (24.8)	643 (22.2)	346 (24.6)
	10.2–10.9	155 (25.6)	794 (27.4)	312 (22.1)
	11.0–12.1	156 (24.1)	682 (23.5)	322 (22.9)
	12.2<	145 (23.9)	781 (26.9)	429 (30.5)
Non-physician healthcare personnel years per 10,000 inhabitants	<213.9	144 (23.8)	583 (20.1)	275 (19.5)
	213.9–258.97	157 (25.9)	681 (23.5)	287 (20.4)
	258.97–311.82	155 (25.6	867 (29.9)	404 (28.7)
	311.82	150 (24.8	769 (26.5)	443 (31.4)
Size of municipality	<5000 inhabitamts	73 (12.1)	310 (10.7)	223 (15.8)
	5000–15,000 inh	116 (19.4)	623 (21.5)	353 (25.1)
	15,000≤ inh	417 (68.8)	1967 (67.8)	833 (58.1)

* Others include unmarried, widowers, divorced or separated and others.

**Table 2 healthcare-12-00943-t002:** Patients use of health service last 365 days of life (number of patients with at least one visit).

Type of Services	Breast Cancer N (% of Total)	Dementia N (% of Total)	Heart Failure N (% of Total)
Hospital admission	388 (64.0)	292 (10.0)	448 (31.8)
Hospital—outpatient or day stays	556 (91.7)	977 (33.7)	870 (61.7)
Nursing homes—long-term stays	102 (16.8)	2453 (84.6)	592 (42.0)
Nursing homes—short term stays	289 (47.7)	547 (18.9)	579 (41.1)
General practise (GP) visits	535 (88.2)	1167 (40.2)	923 (65.5)
Emergency room (local)	296 (48.8)	1322 (45.6)	709 (50.3)
Home nursing	403 (66.5)	741 (25.5)	818 (58.1)

**Table 3 healthcare-12-00943-t003:** Associations between patent characteristics, supply side variables and staying at home the last day before death. Odds ratio (95% Wald Confidence Limits).

		Breast Cancer	Dementia	Heart Failure
Gender	Male	Ref.	Ref.	Ref.
	Female	2.37 (0.41–13.79)	0.91 (0.72–1.17)	0.76 (0.59–0.98)
Age	80–89 years	Ref.	Ref.	Ref.
	<50 years	2.28 (1.06–4.90)	-	6.94 (0.85–56.34)
	50–59 years	2.23 (1.25–3.99)	-	3.62 (1.15–11.40)
	60–69 years	2.94 (1.69–5.11)	0.92 (0.32–2.68)	4.95 (2.32–10.53)
	70–79 years	1.78 (1.07–2.97)	0.84 (0.57–1.23)	1.87 (1.31–2.68)
	90 years≤	1.07 (0.56–2.02)	1.06 (0.85–1.33)	0.63 (0.51–0.86)
Education	Primary	Ref.	Ref.	Ref.
	Secondary	0.83 (0.56–1.25)	1.14 (0.91–1.43)	1.20 (0.94–1.52)
	Higher	1.01 (0.62–1.22)	1.02 (0.73–1.44)	1.43 (0.97–2.12)
Marital status	Others	Ref.	Ref.	Ref.
	Married	1.22 (0.85–1.76)	1.19 (0.92–1.53)	1.12 (0.85–1.48)
Comorbidities	0	Ref.	Ref.	Ref.
	1–2	0.95 (0.64–1.41)	0.84 (0.59–1.20)	0.59 (0.38–0.90)
	3–4	0.69 (0.38–1.22)	0.94 (0.65–1.36)	0.54 (0.35–0.82)
	5 or more	0.78 (0.34–1.80)	1.20 (0.80–1.80)	0.40 (0.26–0.63)
General Practisioners (GPS)s per 10,000 inhabitants	<10.1	Ref.	Ref.	Ref.
	10.2–10.9	1.51 (0.93–2.44)	0.57 (0.41–0.78)	0.97 (0.70–1.36)
	11.0–12.1	1.68 (1.00–2.80)	0.85 (0.64–1.14)	0.84 (0.60–1.18)
	12.2<	2.13 (1.17–2.28)	0.54 (0.38–0.75)	1.04 (0.71–1.51)
Non-physician healthcare personnel years per 10,000 inhabitants	<213.9	Ref.	Ref.	Ref.
	213.9–258.97	0.52 (0.32–0.86)	1.95 (1.36–2.78)	1.08 (0.75–1.55)
	258.97–311.82	0.77 (0.45–1.32)	1.96 (1.37–2.81)	1.37 (0.96–1.97)
	311.82	0.69 (0.38–1.28)	1.48 (0.98–2.23)	1.13 (0.74–1.72)
Population size	<5000	Ref.	Ref.	Ref.
	5000–14,900	1.16 (0.59–2.28)	1.31 (0.89–1.94)	1.16 (0.79–1.70)
	15,000<	1.49 (0.74–3.03)	0.55 (0.36–0.84)	1.12 (0.74–1.70)
N		600	2878	1397
Somer’s D		0.33	0.27	0.32
Percent concordant		66.7	63.4	66.0

## Data Availability

Data from the Norwegian Patient Registry, the Norwegian Registry for Primary Health Care and Statistics Norway have been used in this publication. The interpretation and reporting of these data are the sole responsibility of the authors, and no endorsement by the registry owners is intended nor should be inferred.

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
