# Peer review of "Exploring End-of-Life Care for Patients with Breast Cancer, Dementia or Heart Failure: A Register-Based Study of Individual and Institutional Factors"

_healthcare, 2024, doi:10.3390/healthcare12090943_

Round 1

Reviewer 1 Report

Comments and Suggestions for Authors

Overall, this paper is very well written, and the research is of high quality. I recommend that the authors add the word limitations to line 218 on page 10. I also recommend strengthening the conclusion with a more complete summary, adding the same language as the conclusion in the abstract. 

Look at the reference for # 5 for WHO, line 250 in references. The link brings the reader to the Israel conflict. It is also listed in an incorrect format. 

Author Response

Overall, this paper is very well written, and the research is of high quality. I recommend that the authors add the word limitations to line 218 on page 10.

Answer: Thank you for your suggesstion; we have incorporated the expression.

I also recommend strengthening the conclusion with a more complete summary, adding the same language as the conclusion in the abstract. 

Answer: We have included a conclusion as per you suggestion. Thank you.

Look at the reference for # 5 for WHO, line 250 in references. The link brings the reader to the Israel conflict. It is also listed in an incorrect format. 

Answer: We have reviewed and rectified all references.

Reviewer 2 Report

Comments and Suggestions for Authors

congrats to study the end of life of People with chronic disease.

However the object of study it’s interessant the work has a lot of problems. For example the type of disease can not be comparate because the Numbers of population and the place that it’s normal the person living of the end life,

maybe it’s better just analyse heart failure and breast Cancer.

it’s not clear the kind of care and the expectatives of patients,when they know  that are dieying?

In my opinion it’s necessary to reflect about the work.

Author Response

congrats to study the end of life of People with chronic disease.

However the object of study it’s interessant the work has a lot of problems. For example the type of disease can not be comparate because the Numbers of population and the place that it’s normal the person living of the end life,

maybe it’s better just analyse heart failure and breast Cancer.

it’s not clear the kind of care and the expectatives of patients,when they know  that are dieying?

In my opinion it’s necessary to reflect about the work.

Answer: The topic you raised is indeed fascinating, although it falls outside the scope of our article. Nonetheless, we appreciate the valuable research idea. Thank you.

Reviewer 3 Report

Comments and Suggestions for Authors

Thank you for the opportunity of reviewing the manuscript "Exploring End-of-Life Care for Patients with Breast Cancer, Dementia or Heart Failure: A Register-Based Study of Individual and Institutional Factors".  This is an interesting paper on an important topic. 

Some concrete issues I encountered:

- Abstract: please add more details in section results (I know about the limitations of words but authors should write in this section more the most important information)

- Introduction: I can not find any information about the results of other studies expecially about End-of-Life Care for patients with breast cancer, dementia or heart failure (this section introduces readers to the next stage of the manuscript, the research results)

- Material and Methdods: section Ethical Statement is missing 

- Discussion- authors should add limitations of study, add more results other studies and write implications for practice 

- references- a lot of mistake, for example point 5

Author Response

Thank you for the opportunity of reviewing the manuscript "Exploring End-of-Life Care for Patients with Breast Cancer, Dementia or Heart Failure: A Register-Based Study of Individual and Institutional Factors".  This is an interesting paper on an important topic. 

Some concrete issues I encountered:

- Abstract: please add more details in section results (I know about the limitations of words but authors should write in this section more the most important information)

Answer: Thanks for this suggestion. We have added abetter description of the results and also made the conclusion clearer.

- Introduction: I can not find any information about the results of other studies expecially about End-of-Life Care for patients with breast cancer, dementia or heart failure (this section introduces readers to the next stage of the manuscript, the research results)

Answer: The topic of our article revolved around the utilization of healthcare services among critically ill individuals in their final stages of life, rather than providing an in-depth description of end-of-life care. Therefore, in the introduction, we simply defined what end-of-life care entails and briefly outlined the system of care for these patients in Norway.

- Material and Methdods: section Ethical Statement is missing 

Answer: The Ethical Statement has been included under the section titled "Institutional Review Board Statement."

- Discussion- authors should add limitations of study, add more results other studies and write implications for practice 

Answer: We emphasized the limitations of the study, included additional research in the discussion, and outlined the significance of our research for practical application.

- references- a lot of mistake, for example point 5? 

Answer: We have reviewed and rectified all references.

Reviewer 4 Report

Comments and Suggestions for Authors

Dear Authors, although I am not qualified in the demands of statistical analysis, I read your contribution with great interest. 

I see very interesting outcomes: the importance of the availabitiy of general practitioners (in many regions a problem); the excellent data base for your study; the divergencies between the three groups studied.

I have two remarks:

- your conclusion concerning "advanced care planning" is not really prepared in the contribution (or commented, or documented)....it comes therefore as a little surprise.

- the peculiarity of Norway ( I suppose a country with large distances between the inhabitants) (e.g. in contradiction with many other European countries, mainly Western European countries).

Author Response

Dear Authors, although I am not qualified in the demands of statistical analysis, I read your contribution with great interest. 

I see very interesting outcomes: the importance of the availabitiy of general practitioners (in many regions a problem); the excellent data base for your study; the divergencies between the three groups studied.

I have two remarks:

- your conclusion concerning "advanced care planning" is not really prepared in the contribution (or commented, or documented)....it comes therefore as a little surprise.

Answer: We revised the conclusion regarding our findings. However, we highlighted the potential significance of our data for advance care planning and resource management across different regions, considering the epidemiological situation in each area. This sentence has been incorporated into the discussion. Thank you for your input.

- the peculiarity of Norway ( I suppose a country with large distances between the inhabitants) (e.g. in contradiction with many other European countries, mainly Western European countries).

Answer: We agree with your suggestion. However, our focus was on service utilization in Norway. Further research could explore comparisons with other EU countries—an intriguing prospect for potential collaboration.

Round 2

Reviewer 2 Report

Comments and Suggestions for Authors

Congrats for the authors, text is clearer